# Fact-based Counter Narrative Generation to Combat Hate Speech

## Abstract

Online hatred has become an increasingly pervasive issue, affecting individuals and communities across various digital platforms. To combat hate speech in such platforms, counter narratives (CNs) are regarded as an effective method. In recent years, there has been growing interest in using generative AI tools to construct CNs. However, most of the generative models produce generic responses to hate speech and can hallucinate, reducing their effectiveness. To address the above limitations, we propose a counter narrative generation method that enhances CNs by providing non-aggressive, fact-based narratives with relevant background knowledge from two distinct sources, including a web search module. Furthermore we conduct a comprehensive evaluation using multiple metrics, including LLM-based measures for persuasion, factuality, and informativeness, along with human and traditional NLP evaluations. Our method significantly outperforms baselines, achieving an average factuality score of 0.915, compared to 0.741 and 0.701 for competitive baselines, and performs well in human evaluations.

## CCS Concepts

• **Do Not Use This Code** → **Generate the Correct Terms for Your Paper**; *Generate the Correct Terms for Your Paper*; Generate the Correct Terms for Your Paper; Generate the Correct Terms for Your Paper.

## Keywords

Hate speech, Counter narrative, Fact-based narrative, Large language model

**ACM Reference Format:**
Anonymous Author(s). 2018. Fact-based Counter Narrative Generation to Combat Hate Speech. In *Proceedings of Make sure to enter the correct conference title from your rights confirmation emai (Conference acronym 'XX).* ACM, New York, NY, USA, 14 pages. https://doi.org/XXXXXXX.XXXXXXX

## 1 Introduction

The rise of social media has profoundly reshaped society, revolutionizing communication and allowing people to share knowledge, opinions, and experiences with a global audience almost instantaneously [18, 35]. These platforms have democratized information, giving a voice to individuals who might otherwise remain unheard, and have played a critical role in driving social movements, advocating for marginalized groups, and connecting people across geographic and cultural divides [29, 35]. However, while social

media has brought numerous benefits, it has also given rise to a number of serious social issues. Among the most concerning is the spread of hate speech.

Hate speech thrives in the virtual environment of social media, where anonymity and distance from real-life consequences empower individuals to express harmful, abusive, and often discriminatory rhetoric without fear of accountability [24, 47]. The viral nature of social media allows toxic messages to spread rapidly, reaching vast audiences very quickly [21]. What might have once been an isolated comment in a small community can now go viral, exposing thousands or even millions of people to hateful rhetoric. The ease with which this content spreads and its broader reach allows hate speech to gain momentum quickly and become ingrained within online communities thereby normalizing abusive behavior and making it increasingly difficult to address.

**Widespread effects of online hate speech.** The surge in online abuse and hate poses a complex and pervasive threat to societal security. Victims of hate speech frequently experience profound psychological harm, such as anxiety, depression, and social isolation [1, 40]. Continuous exposure to toxic rhetoric erodes a person's sense of safety and self-worth, potentially leading to long-lasting mental health issues. This impact is particularly devastating for marginalized groups, who are often disproportionately targeted by online abuse. In these cases, hate speech exacerbates existing social inequalities, deepening the psychological toll. The repercussions of online hate are not confined to the digital space. Online hate speech often serves as a precursor to real-world violence, as extremist ideologies and hateful rhetoric spread across social media, inciting or inspiring acts of physical aggression [22, 25, 26]. The unchecked proliferation of hate speech on social media platforms threatens not only individuals but the broader fabric of society. It fosters division, fuels fear, and undermines the mutual respect and understanding essential for maintaining social cohesion.

**Counter narrative as promising combat strategy.** In response to the increasing prevalence of online hate, counter narrative has emerged as a proactive and promising strategy [4, 33]. Unlike traditional methods such as content moderation, which primarily focus on removing harmful material post-dissemination, counter narrative strategy takes a more dynamic approach by preventing the spread of hate while also safeguarding free speech [4, 17, 33]. This strategy involves directly confronting harmful narratives with positive counterarguments, empowering individuals and communities to challenge toxic content as it arises. By interrupting the momentum of hate, counter narrative fosters more inclusive online discussions. Beyond simply mitigating the damage caused by online hate, this approach encourages a culture of empathy, respect, and constructive communication, facilitating a shift towards more positive interactions in digital spaces.

The existing literature on counter narrative generation faces several limitations. Despite several attempts of counter narrative generation, obtaining high-quality, effective and factual counter narrative remains a significant challenge. Only a few methods focus

**Table 1: Comparison with previous works across various criteria of counter narrative generation. 'Yes' ('No') suggests that the particular criterion is present (absent) in the paper. The table suggests that TKGCN [7] as a competitive baseline. Our experiments (Section 4) show that our method outperforms TKGCN [7] in almost all settings.**

| Paper | New Dataset | External Knowledge | | Inject Facts | Eval. with Different LLMs | Group Eval. | Human Eval. |
|---|---|---|---|---|---|---|---|
| | | Knowledge Repo. | Web Search | | | | |
| MTCo[10] | Yes | No | No | No | No | No | Yes |
| RAUCG [15] | No | Yes | No | Yes | No | Yes | Yes |
| TKGCN [7] | Yes | Yes | No | Yes | Yes | Yes | Yes |
| **Ours** | Yes | Yes | Yes | Yes | Yes | Yes | Yes |

on the factual accuracy of counter narratives, which is crucial for effectively exposing the flaws in hate speech. This poses a particular issue as many state-of-the-art techniques rely on pre-trained LLMs, which are prone to generating hallucinated or inaccurate information. These factual inconsistencies can compromise the credibility of counter narratives and foster mistrust, reducing its impact. Additionally, there is a notable gap in the thorough evaluation of generated counter narratives, with limited attention given to assessing its factual accuracy, relevance, and overall efficacy in combating hate speech.

**Our contributions.** To address these limitations, we propose a novel method for generating fact-based counter narratives. Table 1 shows the major differences with the existing works. Our main contributions are as follows.

- We improve the effectiveness of counter narratives (CNs) generated by large language models (LLMs), which often suffer from being generic. To enhance the quality of CNs, we propose a framework that responds with non-aggressive and fact-based feedback, by incorporating relevant background knowledge from two distinct sources.
- We conduct a comprehensive evaluation of the counter narratives using multiple metrics. We use LLM based metrics that measure the persuasion, factuality, and informativeness of our CNs, as well as target group-wise evaluation, human evaluation and traditional NLP metrics.
- Our method significantly outperforms the baselines across several measures. Our generated CNs achieve an average factuality score of 0.915, compared to the two competitive baselines achieving just 0.741 and 0.701. Additionally, our generated CNs also perform well in human evaluation.

**Code.** The codebase is available at the following link: https://anonymous.4open.science/r/counternarratives-DC2A/

## 2 Related Work

There has been a growing body of research focused on the creation and automatic generation of counter narrative data to combat online hate speech. Several studies have aimed at building large datasets that consist of counter narrative responses, enabling the development of automated systems for generating effective counter narrative towards hatespeech.

### 2.1 Counter Narrative Dataset Creation

Numerous efforts have been made to create counter-narrative data. Mathew et al. [23] introduced the first dataset, which was created

by annotating YouTube comments to identify counter narrative in response to hate speech. Qian et al. [31] published two large-scale hate speech intervention datasets. These datasets include conversations collected from social media platforms like Reddit[1] and Gab[2], labeled as hate speeches, with 40K intervention responses written by 900 Mechanical Turk workers. Recognizing that the meaning of certain statements can change depending on context, Yu et al. [41] examined the role of conversational contexts in the annotation and detection of hate and counter speeches, releasing a context-aware dataset. Hundreds of non-expert annotators were invited to label Reddit comments with and without context as hate speech, counter narrative, or neutral speech. Chung et al. [6] introduced the first high-quality multilingual counter narrative dataset, CONAN, which contains around 6K pairs of hate speech and counter narratives in English, French, and Italian. This dataset was meticulously curated by over 100 well-trained NGO experts, requiring more than 500 person-hours to complete.

To reduce the manual labor of generating data, Tekiroğlu et al. [37] proposed an author-reviewer framework in which GPT-2 functions as the "author", generating initial counter narrative responses. Experts then act as "reviewers", tasked with filtering, refining, and post-editing these machine-generated responses. Fanton et al. [10] proposed a human-in-the-loop data collection method where experts post-edit generated counter narratives, iteratively refining the dataset to enhance the generative model's output. Over 18 weeks, this method resulted in 5,000 pairs of hate speech and counter narratives. Bonaldi et al. [5] also utilized this author-reviewer pipeline but focused specifically on collecting counter narratives in multi-turn dialogues, further enhancing the quality and scope of the dataset.

### 2.2 Counter Narrative Generation

Several efforts have focused on generating counter narratives using Natural Language Generation (NLG) techniques.

Qian et al. [31] evaluated the performance of various basic generative models, such as sequence-to-sequence models, for generating counter narratives and found that they often lacked relevance and variety, with many responses being generic or irrelevant. To address these limitations, Zhu et al. [46] proposed a three-module pipeline called Generate, Prune, Select (GPS), which enhances both data diversity and relevance. The method generates diverse candidate samples and incorporates a retrieval-based selection mechanism

---

[1]https://www.reddit.com/
[2]https://gab.com/

to refine them. Chung et al. [7] integrated generative models with information retrieval techniques, leveraging external knowledge to enrich counter narratives and reduce hallucinations—situations where the model generates inaccurate or fabricated information. Jiang et al. [16] utilized stance consistency, semantic overlap, and relevance to hate speech in constructing a knowledge repository from the ChangeMyView subreddit [3]. This repository is then employed to provide knowledge-augmented counter narratives.

Several works have explored additional stylistic aspects of counter narratives to improve its effectiveness. To enhance specificity, Bonaldi et al. [5] applied two attention-based regularization techniques, incorporating a broader context during both the training and generation phases. Furman et al. [11] emphasized the argumentative structure within hate speech to guide the generation of responses, steering them towards specific counterstrategies. Saha et al. [32] focused on controlling multiple stylistic dimensions simultaneously, including politeness, detoxification, and emotional tone in the generated counter narratives. Gupta et al. [13] proposed a two-stage framework that conditions counter narrative generation on five different strategies: informative, denouncing, questioning, positive, and humorous approaches. These methods allow for more nuanced and contextually appropriate counter narratives, tailored to the nature of the hate speech being addressed.

The most common approach for generating counter narratives involves fine-tuning a pre-trained language model on a dedicated counter narrative dataset in a relatively low-computation setup (e.g., [14, 31, 36]). However, recent advances have enabled more flexible methods, such as generating counter narratives using few-shot learning techniques [3, 8, 11, 38]. Additionally, one-shot and zero-shot prompting have emerged as viable alternatives, allowing counter narrative generation with minimal training data [27, 44]. While these approaches significantly reduce the need for large-scale annotated datasets, making the generation process more accessible, the most of the counter narratives are not fact-based. *In this paper, we build a method to produce counter narrative that are fact-based and thus, more effective.*

## 3 Our Method

Large language models (LLMs) often generate generic or repetitive counter narratives (CNs), reducing their effectiveness in combating hate speech [9]. A more effective approach is to respond with non-aggressive, fact-based narratives that incorporates relevant background knowledge. This not only increases the relevance of the counter narrative to the specific hate speech but also enhances diversity by avoiding repetition [2]. Our proposed framework addresses this by integrating three key components: *(i) document retrieval, (ii) document summarization, and (iii) counter narrative generation.* Figure 1 illustrates the workflow of these steps. We discuss each component in detail in the following subsections.

### 3.1 Document Retrieval

The first step involves retrieving relevant documents, which are subsequently used to craft fact-based CNs. Towards this objective, we generate targeted queries via LLM prompts.

---

[3]https://www.reddit.com/r/changemyview/

*3.1.1 Query Generation.* This step focuses on generating queries to be used in the document retrieval task. We employ step-back prompting [43] to generate queries, where the model is prompted to first consider what information is necessary to construct a response. In our setting, it is crucial that the base LLM understands the goal of the query. For every instance of hate speech, we generate three specific queries denoted as $Q = \{Q_1, Q_2, Q_3\}$ to help us retrieve the related documents containing facts and statistics to counter the hate speech. We use GPT4o[4] with the following prompt. Instead of using key phrases, we include the full instance of the hate speech to ensure that the model becomes aware of the entire context.

> **Prompt for Query Generation**
>
> You are responding with counter speech to the hate speech {hatespeech}.
> Generate 3 precise queries you would research in order to generate counterspeech for this hate speech.

*3.1.2 Document Retrieval.* Next, we aim to retrieve relevant documents from a knowledge repository. In this step, we use the Newsroom dataset[12] as our primary knowledge base, which includes 1.3 million articles from 38 different news publications. However, a limitation of the Newsroom dataset is that it only includes data from 1998 to 2017. To effectively combat hate speech with current information, it is crucial to integrate more recent datasets. Since facts and statistics are constantly evolving, relying solely on outdated data risks producing ineffective responses. To address this, we also incorporate the Tavily[5] online web search API to retrieve recent documents. This is especially designed for RAG (Retrieval-Augmented Generation) systems [19] and LLMs. It retrieves the most relevant information from the sources based on a query. Thus, we ensure that our generated counter narratives are grounded with the recent information.

To manage and access large amounts of data, we depend on the low dimensional representations or vector embeddings space. By transforming both our knowledge base and queries into vector embeddings, we can perform similarity-based search to find the most relevant documents for each query. We use Chroma[6] to store and retrieve vector embeddings. First, we generate a vector store from our knowledge base. Then, for each query, it is converted to vector embeddings, which are used to retrieve relevant documents from the knowledge repository. Chroma uses *hierarchical navigable small world (HNSW)* [20] as an approximate nearest neighbor search algorithm. In this algorithm, documents are indexed into a hierarchical structure of graphs and edges. Connection between documents are based on their similarity, such as euclidean distance, cosine similarity, or inner product. Given a query, the algorithm starts the search at the top layer to find the nearest neighbor and go into deeper layer if the the similarity is not sufficient. For each instance of hatespeech, we generate 3 queries as described in Section 3.1.1. We retrieve two documents per query from both Newsroom and online search which produces a total of 6 documents. The retrieved

---

[4]https://openai.com/index/hello-gpt-4o/
[5]https://tavily.com/
[6]https://www.trychroma.com/

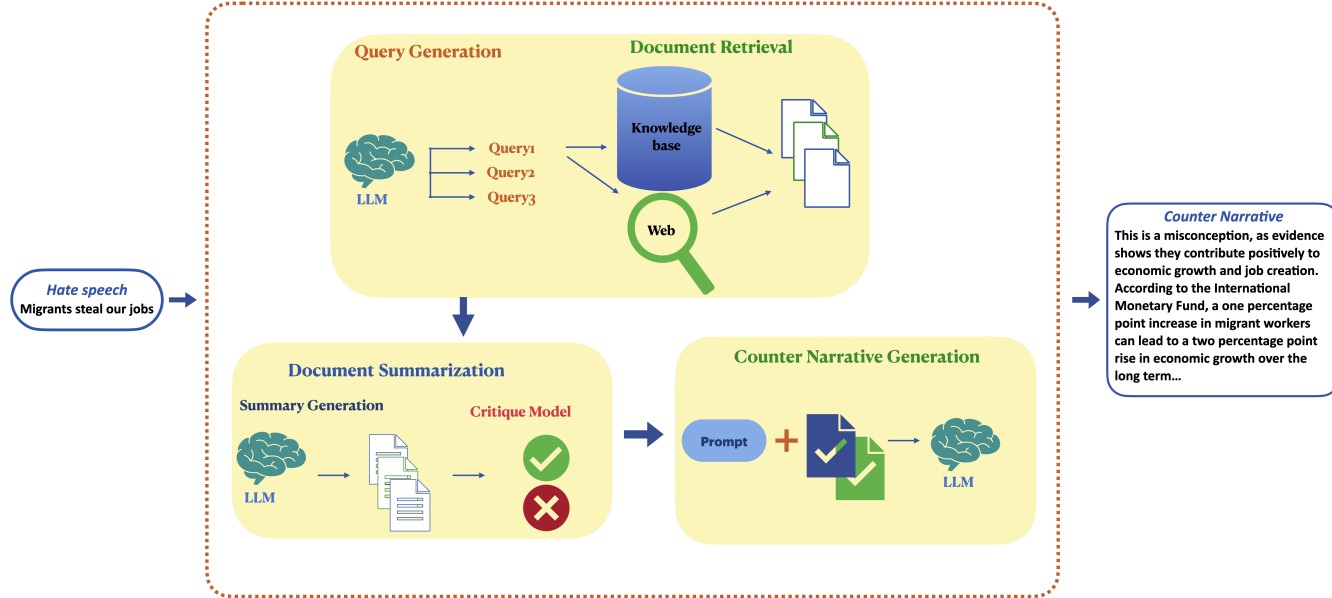

**Figure 1: Our pipeline for generating counter narratives involves three key stages: (1) Query Generation and Document Retrieval, where three queries are generated to retrieve relevant documents from a knowledge base and web search; (2) Document Summarization, where the retrieved documents are summarized, and a critique model determines whether they are relevant; and (3) Counter Narrative Generation, where the relevant documents are incorporated into a prompt to generate the final counter narrative.**

documents are denoted as $D = \{D_{1N}, D_{1W}, D_{2N}, D_{2W}, D_{3N}, D_{3W}\}$ where, for the $i^{\text{th}}$ query, $D_{iN}$ and $D_{iW}$ refer to the documents from Newsroom and the web respectively.

## 3.2 Document Summarization

We obtain relevant documents in the previous step (Section 3.1.2). The challenge with these retrieved documents is that they are lengthy, and can sometimes be irrelevant and noisy. Before injecting information that might not accurately align with the hate speech and subsequent CNs, we need to verify the relevance. So, we split this process into two steps: first, we apply techniques to summarize the key points from the relevant documents, and then we use the critique model to filter and select the relevant documents.

*3.2.1 Summary Generation.* The documents in the retrieved set $D$ are usually lengthy. Besides, LLMs also have a certain limit for tokens that are used for prompts. Additionally, not all paragraphs in the documents contain relevant facts needed for our generation task. To address this, we generate summaries focusing on key facts, and statistics from the documents. We employ GPT-4o to summarize the contents as it has shown a great potential in generating automated text summarization in various domain such as food summary[34], discharge summary for a patient [39], news summary[28], etc. We use the following prompt given below. This process produces the set of summarized documents $D_S = \{D_{1NS}, D_{1WS}, D_{2NS}, D_{2WS}, D_{3NS}, D_{3WS}\}$ where $D_{iNS}$ refers to the summarized document from Newsroom and $D_{iWS}$ refers to the document summary retrieved from the web for the $i^{\text{th}}$ query.

> **Prompt for Summary Generation**
>
> Summarize this document in 2-3 sentences in a way that answers the query and retain information relevant for countering the hate speech. Keep any relevant statistics, and sources.
> Query: {query}
> Hate speech: {hs}
> Document: {document}

*3.2.2 Critique Model.* As the critique model, we utilize GPT-4o with a specific prompt. The prompt includes the query used to generate the documents, the retrieved documents, the identified hate speech, and the purpose of the documents, which is to counter the hate speech. The output of the model (GPT-4o) for a document is binary with "yes" or "no". The detailed prompt used for this process is provided in Appendix C. After processing with the critique model, we derive a subset of relevant documents $D_C$ where the final document set can be of any number between one to six. Generally speaking, the critique model is used for filtering the irrelevant ones. Now there is a possibility that the critique model might yield an empty set, indicating no relevant documents have been found. However, after testing with multiple instances of hate speech, we observe that this scenario is rare. The critique model always returns at least one relevant document for the task in all cases.

## 3.3 Counter Narrative Generation

To effectively counter hate speech, our final step involves generating counter narratives that directly address the underlying misinformation, prejudice, or harmful rhetoric present in the hate speech instance. These counter narratives aim to promote fact-based corrections. For this generation process, we employ several LLMs including GPT-4o[7], GPT-3.5[8], Gemini 1.5 Flash[9], Claude Sonnet 3.5[10], and Mistral 7B[11]. The detailed prompt is provided in Appendix C. The task of the model is to generate a counter narrative that is both factually accurate and aligned with the goal of promoting positive, alternative perspectives. The prompt $p = \{h_{si}, D_{Ci}, instruct\}$ includes the hate speech instance $h_{si}$, the summarized documents $D_{Ci}$ and instructions for generating a counter narrative of $i^{th}$ instance of the dataset. After running evaluations based on toxicity, factuality, persuasiveness, and informativeness as shown in Table 2, we select GPT-4o as the preferred model. It consistently produces non-toxic and coherent counter narratives, outperforming the other models in our trial. Examples of hatespeech and counter narratives generated by MTCo [10], TKGCN [7], and ours are shown below. Note that, counter narratives generated by our model has more facts and it cites the sources (shown in green).

---

**Example of Hate Speech (HS) & Counter Narratives (CNs)**

**HS**: Gays are spreading AIDS all around the world because they are promiscuous and fornicate with strangers, we should prevent them to make blood donations.

**CN (MTCo)**: We can simply disallow donations from those who had unprotected sex or a high number of partners, instead of making sexual orientation the basis as in your claim.

**CN (TKGCN)**: The real issue is unprotected sex. Regardless of sexual orientation, it is dangerous for anyone to have unprotected sex that could lead to a life threatening disease.

**CN (Ours)** : The notion that gay individuals are responsible for spreading AIDS is a harmful stereotype not supported by scientific evidence. Studies have shown that HIV transmission can be effectively prevented when the HIV-positive partner is on antiretroviral treatment, regardless of sexual orientation (source: https://www.ncbi.nlm.nih.gov/pmc/articles/PMC8063169/). Furthermore, the Red Cross implements rigorous safety protocols to ensure that all eligible individuals, regardless of sexual orientation, can safely donate blood (source: https://www.redcrossblood.org/donate-blood/dlp/share-your-health.html). Discriminatory practices based on sexual orientation hinder efforts to combat HIV/AIDS and ignore the fact that transmission is related to specific behaviors, not sexual orientation itself.

---

[7] https://openai.com/index/hello-gpt-4o/
[8] https://platform.openai.com/docs/models/gpt-3-5-turbo
[9] https://deepmind.google/technologies/gemini/flash/
[10] https://docs.anthropic.com/en/docs/welcome
[11] https://mistral.ai/news/announcing-mistral-7b/

## 4 Experimental Results

In this section, we discuss the experimental set up and provide results on various measures. More specifically,

- We use various evaluation metrics to validate our counter narrative (CNs) generations in terms of NLP measures, fact-based measures, and linguistic quality. We also perform human evaluation of our generated CNs.
- Since our pipeline consists of multiple stages or components, we assess the quality of each step individually. Furthermore, we evaluate the CNs for individual target group. Finally, we make some interesting observations between the original hate speech and our constructed CNs.

### 4.1 Set up

**Datasets.** We use the Multi-Target CONAN [10] dataset for our analysis. This dataset has hate speech and counter narrative pairs in English, targeting multiple groups. It has been collected using a Human-in-the-Loop approach to ensure quality and relevance. The dataset contains 5,003 hate speech and counter narrative pairs, addressing various hate targets such as people with disabilities, Jews, LGBT+ , Muslims, migrants, people of color (POC), and women. A few examples are shown in Appendix A.

**Baselines.** We use the Multi-Target CONAN [10] dataset as our first baseline. This paper proposes a human-in-the-loop approach to generate counter narratives. As seen in the examples in Table 7, the CNs produced by this method are not fact-based. We also compare our results with TKGCN [7], where counter narratives are similarly grounded in knowledge from a repository. However, our method outperforms this baseline in almost all settings. Another relevant work, RAUCG [15] has not made the code and data publicly available, and therefore we have not included this baseline for our experiments.

### 4.2 Evaluation of Counter Narrative Generation on Different Measures

We evaluate the quality of our counter narrative generation with various measures in this experiment. Table 2 presents all the results along with two main baselines: TKGCN[7] and MTCo [10]. We also show variations with different LLMs for the last step where we generate the counter narratives (CNs).

We would like to emphasize the fact that it is challenging to evaluate the CNs because of the lack of ground truth. Thus, we evaluate these from different viewpoints and measures that are practical for CNs. We describe each measure along with results in details as follows.

**Toxicity.** Toxicity in hate speech is expected to be high. In contrary, toxicity does not help in counter narrative (CN) for obvious reasons. Our objective is to ensure that our created CN exhibits lower toxicity. To validate this, we utilize the Perspective API[12] developed by Jigsaw and Google. This is based on the BERT-based model and Convolutional Neural Networks (CNNs). For a given text, it returns a probability score ranging from 0 to 1, where 0 indicates the text is non-toxic, and 1 means it is highly likely to be toxic.

[12] https://perspectiveapi.com/

**Table 2: Evaluation of the counter narratives with different measures. We compare our proposed model with two main baselines TKGCN [7] and MTCo [10]. We also show variations with different LLMs for the last step where we generate the counter narratives. "Tox.", "Fact.", "Pers.", "Inf.", and "LQ" refer to toxicity, factuality, persuasiveness, informativeness, and linguistic quality, respectively. For toxicity, lower values indicate better performance, whereas for the other metrics higher values are better. The best scores in each metric are highlighted in bold. For BLEU and BERTScore we compare all the CNs with the given CNs from MTCo [10] Dataset. Our proposed method outperforms others especially in terms of factuality, persuasiveness, and informativeness.**

|  | Model | Tox. | | Fact. | Pers. | Inf. | LQ | BLEU | BERTScore |
|---|---|---|---|---|---|---|---|---|---|
|  |  | HS | CN |  |  |  |  |  |  |
| **Baselines** | **MTCo [10]** | 0.501 | 0.161 | 0.701 | 0.620 | 0.661 | 0.791 | – | – |
|  | **TKGCN [7]** | 0.501 | 0.201 | 0.741 | 0.673 | 0.695 | 0.845 | **0.046** | 0.847 |
| **Our Methods** | **Gemini** | 0.501 | 0.145 | 0.892 | 0.845 | 0.808 | 0.849 | 0.030 | 0.847 |
|  | **Claude** | 0.501 | 0.159 | 0.905 | 0.858 | 0.874 | 0.854 | 0.024 | 0.840 |
|  | **Mistral** | 0.501 | 0.125 | 0.866 | 0.824 | 0.764 | 0.831 | 0.036 | 0.850 |
|  | **GPT3.5** | 0.501 | **0.094** | 0.866 | 0.820 | 0.773 | **0.859** | 0.034 | **0.851** |
| **Our Main Model** | **GPT4o** | 0.501 | 0.131 | **0.915** | **0.859** | **0.879** | 0.832 | 0.02 | 0.827 |

In Table 2, we present the toxicity levels of both hate speech and generated counter narratives using different LLMs. The hatespeech has a toxicity score of 0.50, where the lowest toxicity score for the CN is 0.094, produced by GPT3.5. The highest score, 0.20, comes from the counter narratives proposed by [7]. Our proposed model achieves a score of 0.13 which is a significant improvement over the existing works.

**Factuality, Persuasiveness, and Informativeness.** This is the most significant experiment in our case. We follow the method used by [15] to measure factuality, persuasiveness and informativeness. We use GPT 3.5 to score these three criteria in the range of 0 to 1. In our case, we would like to have higher scores in all the three metrics as our objective is to generate CNs that are informative, factually reliable and most importantly persuasive. Here, persuasiveness means how the CNs are effective to influence the readers opinion. The results in Table 2 show that our CNs receive the highest scores across all three metrics: 0.91 for factuality, 0.85 for persuasiveness, and 0.87 for informativeness. In comparison, the method in [7] produces lower scores such as 0.74, 0.67, and 0.69 for these metrics respectively.

**Linguistic Quality.** Most NLP metrics (e.g., BLEU, BERTScore) rely on reference texts to evaluate generated text. GRUEN [45], however, is a reference-less linguistic quality metric that assesses text quality based on several aspects: grammar, non-redundancy, focus, and coherence. It evaluates whether the text is grammatically correct, free of unnecessary repetition, maintains topic relevance within paragraphs, and flows coherently between sentences. GRUEN provides a holistic quality score. Table 2 shows that our approach with GPT-3.5 achieves the highest score (0.85) where as our main model produces 0.83. The slight difference may be due to the fact that, GPT-3.5 does not include any source while injecting facts to the CNs where our main model does the same with relevant sources. We also provide examples in Table 8 and Table 9 in Appendix B.

**BLEU-2.** Besides the above measures we also aim to evaluate the similarity of our CNs with the original CNs in the Multi-Target CONAN [10] dataset. The BLEU [30] score is commonly used to

assess the quality of machine-generated text by comparing them to reference text. BLEU-2 focuses on 2-gram precision, meaning it evaluates how well pairs of consecutive words (bigrams) in the candidate text match those in the reference. To compare the CN generation, we chose Multitarget-Conan [10] as our reference text. The score ranges from 0 to 1, with higher scores indicating a closer alignment with the reference text. As shown in Table 2, our CNs do not exhibit significant bigram overlap with reference CNs. This is expected as our CNs incorporate a moderate number of factual statements, which naturally reduces bigram similarity but enhances the informative value of the generated CNs. In comparison, the baseline TKGCN is less factual and shows higher bigram similarity.

**BERTScore.** Similar to the above measure, BERTScore [42] also measures similarity between texts. However, it captures the semantic similarity between the generated text and reference texts by utilizing contextual embeddings. This evaluation focuses on how well the meanings of the generated text align with the reference text—which is Multitarget-Conan [10] data in our case—rather than relying solely on exact word matches. This metric helps us to validate that our counter narratives (CN) maintain a level of semantic similarity, even though the inclusion of factual content may cause slight deviations from the reference. As shown in Table 2, all models achieve relatively similar scores, though our method produces the lowest one as our CNs aim to have other components such as factuality.

## 4.3 Human Evaluation

Human evaluation is widely regarded as the gold standard for assessing the quality of text generation especially by generative models such as LLMs. To validate our counter narratives (CNs), we have randomly chosen three samples and asked 17 researchers to evaluate CNs based on the followings.

- **Factuality**: Do the counter narratives include facts and statistics?
- **Coherence**: Are the counter narratives logically structured and easy to understand?

- **Effectiveness**: Is the counter narrative likely to be persuasive and change someone's perspective on the issue?
- **Relevance**: Are the facts and information presented in the counter narrative directly related to the content of the hate speech?

The users are asked to rate each counter narrative on a scale of 1 to 5, with 5 indicating excellent performance and 1 indicating poor performance. Figure 2a shows the results of human evaluation. In every category, the output are more than 3.5 out of 5 except for *effectiveness*. The slightly lower score in this category may indicate that while the counter narratives are coherent and factual, the users—who evaluate these—think that (or are unsure about whether) these counter narrative may not always have an effect on shifting opinions or persuading the audience.

## 4.4 Component-wise Evaluation

Our framework has multiple steps before the final counter narrative generation. In this experiments, we evaluate the quality of the individual steps.

*4.4.1 Document Retrieval.* In order to inject facts, our framework has a document retrieval component that obtains six documents per hate speech, summarizes them, and assesses their relevance using a critic model. The critic model filters out the summaries that are irrelevant for countering hate speech, ensuring only pertinent information is utilized in the counter narrative generation. In Table 3, we present the average amount of summaries that are irrelevant per hate speech. The low numbers indicate that the documents retrieved are mostly relevant as low amount of filtering is needed. The low scores across each target group show that the retrieval step consistently retrieves relevant documents.

**Table 3: Number of document summaries filtered out per target group. For each hate speech, six documents are retrieved and summarized. Lower scores of filtering here suggest that the documents retrieved are indeed relevant to counter the hate speech.**

| Target | Summaries Filtered |
|---|---|
| DISABLED | 0.074 |
| JEWS | 0.144 |
| LGBT+ | 0.175 |
| MIGRANTS | 0.156 |
| MUSLIMS | 0.106 |
| POC | 0.097 |
| WOMEN | 0.045 |
| OTHER | 0.156 |

*4.4.2 Summarization.* In our framework, the document summarization is an important step. In this experiment, we aim to capture the similarity between the original document and its corresponding generated summary. In Figure 2b and 2c, we present the BERTScore and ROUGE-L between every document retrieved and its summary. The scores are relatively low for ROUGE-L, showing little n-gram overlap due to paraphrasing or alternative wording. However, BERTScore performs generally well, showing that the core

semantic meaning is still retained in our summaries. This means our summarization approach effectively captures the main ideas while expressing them in different words.

## 4.5 Quantitative Analysis for Target Groups

This experiment aims to analyze the counter narratives in each target group. In Table 4, we show a detailed comparison of the CNs generated by our model against those from the Multitarget-Conan [10] dataset. This dataset [10] has eight target groups, and here, we present target-group wise measures based on them. The metrics include the average number of words and sentences, factuality scores, and toxicity scores. Our model consistently generates longer CNs, as measures by the higher word and sentence counts across all target groups. It shows that our model includes more detailed and comprehensive responses. Additionally, our model achieves higher factuality scores which implies its ability to produce fact-based CNs. In terms of toxicity, our model generally has lower or similar scores compared to the MTCo dataset. This shows that even though our CNs are longer and more detailed, they do not become more harmful or hateful.

## 4.6 Interesting Observations

*4.6.1 Are the generated CNs generic?* One of the limitations of the existing counter narrative methods is that the produced counter narratives tend to be generic, which makes them less effective. We aim to show that the CNs—produced by our method—are unique and specifically tailored to each instance of hate speech. In Table 5, we present the BLEU, ROUGE-L, and BERTScore between one CN and every hate speech in a specific target group, and then average those scores. We repeat the process for each data point and report the average score. The results show that both the baselines— MTCo [10] and TKGCN [7]—have higher similarity between the CN and hate speech, whereas our model has less similarity. Since BERTScore accounts for contextual similarity, it still produces a high score, though it is the lowest compared to the other works.

**Table 5: BLEU, ROUGE-L, and BERTScore evaluation between one CN compared with every HS in the target group. Lower scores (best are in bold) in BLEU and ROUGE-L suggest that the CNs are less generic and less directly aligned with the hate speech within each category. BERTScore measures the semantic similarity of the CNs to the reference texts.**

| Model | BLEU | ROUGE-L | BERTscore |
|---|---|---|---|
| **TKGCN [7]** | 0.0064 | 0.0826 | 0.8406 |
| **MTCo [10]** | 0.0100 | 0.0964 | 0.8560 |
| **Ours** | **0.0037** | **0.0516** | **0.8226** |

*4.6.2 How similar are the HS and the corresponding CN?.* To answer this, we also present the BLEU, ROUGE-L, and BERTScore values between every pair of counter narratives (CNs) and hate speech in Table 6. The scores are notably low which shows that the CNs are not simply repeating words or n-grams from the hate speech. It

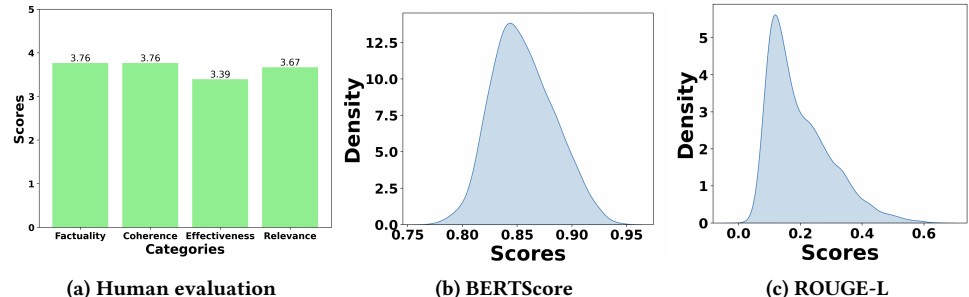

(a) Human evaluation      (b) BERTScore      (c) ROUGE-L

Figure 2: (a) Results of human evaluation across four categories: Factuality, Coherence, Effectiveness, and Relevance. In most of them the output are more than 3.5 out of 5, indicating the high quality of the generated counter narratives. (b-c) Comparison of BERTScore scores and ROUGE-L scores between documents and summaries. Low scores for ROUGE-L suggest that the summaries do not retain similar words as the original documents. However, the summaries retain the semantic meaning of the original documents as indicated by the high BERTScores.

Table 4: Target group-wise evaluation of counter narratives. This table shows the average number of words, sentences, factuality score, and toxicity score for each target group in both our CNs (that our method produces) and the original CNs in the Multitarget-Conan (MTCo) dataset. The best results on factuality and toxicity are in bold. We observe higher word and sentence counts in our CN as it includes factual information. In most of the target groups, our produced CNs achieve lower (better) toxicity scores.

| Target | Words | | Sentences | | Factuality | | Toxicity | |
|---|---|---|---|---|---|---|---|---|
| | Ours | MTCo | Ours | MTCo | Ours | MTCo | Ours | MTCo |
| DISABLED | 113.77 | 27.81 | 2.67 | 1.63 | **0.909** | 0.700 | **0.060** | 0.088 |
| JEWS | 125.55 | 27.96 | 4.57 | 2.10 | **0.959** | 0.644 | 0.229 | **0.204** |
| LGBT+ | 114.60 | 26.23 | 3.68 | 1.80 | **0.924** | 0.650 | **0.117** | 0.217 |
| MIGRANTS | 117.42 | 29.17 | 5.34 | 2.24 | **0.902** | 0.688 | **0.083** | 0.084 |
| MUSLIMS | 122.91 | 23.81 | 4.71 | 1.74 | **0.901** | 0.675 | **0.112** | 0.178 |
| POC | 117.34 | 27.08 | 3.94 | 2.00 | **0.928** | 0.731 | 0.207 | **0.201** |
| WOMEN | 119.60 | 28.70 | 4.54 | 2.00 | **0.900** | 0.708 | **0.121** | 0.162 |
| OTHER | 115.14 | 23.91 | 4.15 | 1.50 | **0.909** | 0.734 | **0.102** | 0.129 |

also validates that our model generates more diverse and context-specific counter narratives, rather than relying on direct overlaps with the original text.

Table 6: BLEU, ROUGE-L, and BERTScore evaluation between HS and CN. The low scores (best results are in bold) in our CNs indicate that the words from the HS are not retained in our CNs, validating that our model better constructs more diverse CNs.

| Model | BLEU | ROUGE-L | BERTscore |
|---|---|---|---|
| TKGCN [7] | 0.0468 | 0.1275 | 0.8523 |
| MTCo [10] | 0.0719 | 0.1712 | 0.8698 |
| Ours | **0.0268** | **0.0840** | **0.8370** |

## 5 Conclusion

In this paper, we have designed a novel method for generating fact-based counter narratives. By integrating relevant background knowledge from multiple sources and focusing on non-aggressive language and facts, our approach significantly improves the quality and effectiveness of CNs in combating hate speech. Our comprehensive evaluation demonstrates the strength of our method across various metrics, including persuasion, factuality, and informativeness. Both LLM-based assessments and human evaluations show that our approach consistently outperforms competitive baselines. These results underscore the potential of our framework to significantly enhance counter narrative generation and its practical application in addressing online hate speech at scale.

*Limitations and Future Work.* Our work assumes that the hatred on social media is primarily expressed through text. However, it is important to recognize that hate can also appear in image form, which remains a significant challenge to address. Additionally, future research could explore the creation of multi-response counter-narratives. When a counter-narrative elicits a reaction from other users, responding appropriately and understanding the dynamics of the conversation will be crucial.

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

## A  Dataset Description

In Table 7, we show target group-wise examples of the dataset Multi target Conan [10].

## B  Examples

Table 8 shows the overall pipeline, including the generated query, retrieved documents, critique model, document summary, and counter-narrative generation process. In Table 9, we present the counter-narratives generated using different LLMs.

## C  Prompts

From query generation to counter narrative generation, we utilize LLMs to produce outputs. We experimented with multiple prompts, and in this section, we presents the prompts that yield the best results. The instructions are written in black, additional information or contexts highlighted in red. Number or sentence specifications are marked in blue.

---

**Prompt for Query Generation**

You are responding with counter speech to the hate speech {hatespeech}.
Generate 3 precise queries you would research in order to generate counterspeech for this hate speech.

---

**Prompt for Summary Generation**

Summarize this document in 2-3 sentences in a way that answers the query and retain information relevant for countering the hate speech. Keep any relevant statistics, and sources.
Query: {query}
Hate speech: {hs}
Document: {document}

---

**Prompt for Critique Model**

You are a grader assessing the relevance of a retrieved document for use in constructing counterspeech to hate speech.
Here is the retrieved document:{document}
Here is the hate speech: {hs}
Here is the query used to retrieve that document: {question}
If the document is generally relevant to countering the hate speech, grade it as relevant. Grade it as irrelevant if it is generally irrelevant to countering the hate speech, or if it seems to agree with the hate speech.

---

**Prompt for CN Generation**

Generate counterspeech to this hate speech, utilizing the following information fetched from various sources. Utilize the information to build your narrative, and cite sources when possible.
Context: {documents}
Hate speech: {hs}
Keep your counter speech to 4 sentences.

---

**Table 7: Target groupwise examples from Multi-target Conan dataset**

| Target | HS | CN |
|---|---|---|
| **DISABLED** | Downies may hinder the productivity of other workers because they are not able to interact in a normal way. They should be isolated. | There are various levels of Down syndrome. Some are suitable for working with other people in "normal workplaces". |
| **JEWS** | Global Jewry did not want Brexit and are doing everything they can to sabotage it. | What makes you think that? Do you have any facts to back this up? |
| **LGBT+** | Why don't gay people seek a cure? | Have you ever talked to a homosexual person? It could help you understand many things, including that sexual orientation is fundamental to a person's identity, not a matter of whim or ill health. |
| **MIGRANTS** | Immigrants just don't want to integrate. Let's kick them out! | Immigrants are integrating very well into our society and it seems that they have reached a better level of English than other people. |
| **MUSLIMS** | Apples and oranges are not the same and cultures have never been 'equal'. Muslims are just misogynist! | You can find outdated teaching in all major religions but it doesn't mean most people follow those parts, a similar example from Christianity: Ephesians 5:22, "Wives, submit to you husbands as to the Lord" |
| **POC** | I hate negros: they killed a cop! | So are you blaming all of them? Lets say you are in a group of 10 people, 1 is a killer, does that make you ALL killers? Are you serious? |
| **WOMEN** | A woman's place is with her husband or his family. | There are many cases where the wife works and provides for the family, making a valuable contribution. It is a source of pride for modern men to be married to these women. |
| **OTHER** | Gypsy rape our women. They are criminals. | Forgive me but I don't understand how rape can be a Gypsy characteristic. Rather it is the product of toxic misogynistic culture. |

**Table 8: Example of Generated CN along with all the steps mentioned in our pipeline. As the retrieved documents are lengthy, few lines of the documents are shown. D1N, D1W denotes documents from Newsroom and web search, and S1N, S1N denotes summaries of the documents D1N, and D1W, respectively.**

| | |
|---|---|
| **HS** | Immigrants are overrunning our country, and are here illegally |
| **Generated Query** | **Q1**: What are the economic contributions of immigrants to the country, including both documented and undocumented immigrants? 
 **Q2**: What are the legal pathways and challenges for immigrants seeking to enter and reside in the country? 
 **Q3**: What are the current statistics on immigration rates compared to historical data to assess the claim of overrunning? |
| **Retrieved Documents** | **D1N**: The consensus of the economic analysis is that the taxes paid by immigrants and their descendants exceed the benefits they receive. Even with undocumented immigrants, the Internal Revenue Service estimates that they paid almost $50 billion in federal taxes between 1996 to 2003, including payroll and Social Security... 
 **D1W**: Immigrants make significant contributions to our economy on virtually every front - including on tax revenue, where they contribute $458.7 billion to state, local, and federal taxes in 2018. If the U.S. provided a pathway to permanent residency and citizenship, it would increase Americans' income by $791 billion and generate $184 billion in additional state and local and federal tax revenue from currently undocumented immigrants, and add more than 200,000 jobs to the U.S. economy per year... 
 **D2N**: The immigration bill being drafted by Congress has bipartisan support on three broad concepts - a pathway to citizenship for the undocumented, streamlining legal immigration and more stringent enforcement of the laws against hiring illegal workers... 
 **D2W**: CAP analysis shows that practices and policies that facilitate legal pathways can reduce irregular migration and encourage prospective migrants to seek entry into the United States through such ... |
| **Document Summary** | **S1N**: Immigrants, both documented and undocumented, make significant economic contributions to the U.S. economy. The Internal Revenue Service estimates that undocumented immigrants paid nearly $50 billion in federal taxes from 1996 to 2003, and their purchasing power supports hundreds of thousands of U.S. jobs. Despite some working off the books, the overall economic impact of immigrants is positive, as highlighted by the Council on Foreign Relations, which warns that failing to implement effective immigration policies could weaken the U.S. economy and national security. 
 **S1W**: Immigrants, both documented and undocumented, play a vital role in the U.S. economy, contributing $458.7 billion in taxes in 2018 alone. Providing a pathway to permanent residency and citizenship for undocumented immigrants could further boost the economy by increasing Americans' income by $791 billion, generating an additional $184 billion in tax revenue, and creating over 200,000 jobs annually. These contributions highlight the positive economic impact of immigrants, countering the narrative that they are overrunning the country. 
 **S2N**: The immigration bill being drafted by Congress aims to address three main areas: creating a pathway to citizenship for undocumented immigrants, streamlining the legal immigration process, and enforcing laws against hiring illegal workers. However, the legal immigration system is complex, with multiple pathways and categories, each with limited visas and specific criteria, leading to significant backlogs. For instance, a Migration Policy Institute study noted that as of November 2012, over 4.4 million people with approved visa petitions were awaiting processing, primarily from family-based categories, and it could take 19 years to clear these backlogs without additional visas, highlighting the challenges in reforming the system to accommodate both current and new applicants. 
 **S2W**: The CAP analysis highlights that establishing and promoting legal pathways for immigration can significantly reduce irregular migration by encouraging prospective migrants to enter the United States through official channels. Legal pathways include options such as family reunification, employment-based visas, and humanitarian programs, which provide structured and lawful means for immigrants to reside in the country. By expanding and streamlining these processes, the U.S. can effectively manage immigration and counter the misconception that immigrants are predominantly entering the country illegally. |
| **Critique Model** | **S1N**: relevant, **S1W**: relevant, **S2N**: relevant, **S2W**: relevant |
| **Counter Narrative (CN)** | The notion that immigrants are "overrunning" the country is not supported by data, as unauthorized immigrants make up only 28% of the foreign-born population, and states like California, Illinois, and New York have seen declines in their unauthorized populations (Pew report). Immigrants, both documented and undocumented, contribute significantly to the U.S. economy, paying $458.7 billion in taxes in 2018 alone, and their presence is projected to increase Americans' income by $791 billion if given a pathway to citizenship (PDF Immigration Facts). Furthermore, the U.S. net immigration rates show only a modest increase, with 9.27 million net immigrants projected from 2022 to 2024, which is comparable to previous decades (Visual Capitalist). By expanding legal pathways and reforming immigration policies, the U.S. can effectively manage immigration and harness the positive economic impact immigrants bring (CAP Analysis). |

**Table 9: Example of Generated CN using different LLMs**

| | |
|---|---|
| **HS** | Allowing gays to donate blood compromises patient safety because they are having unprotected anal sex all the time. |
| **Gemini** | This is a harmful and unfounded generalization. The FDA's new blood donation guidelines focus on individual risk assessment, not blanket bans based on sexual orientation. Countries like Britain and Italy have successfully implemented individual risk assessment for all donors, demonstrating that allowing gay individuals to donate blood does not compromise safety. It's important to remember that sexual behavior is diverse and not limited to any one group, and making assumptions about individuals based on stereotypes is harmful and inaccurate. |
| **Claude** | This harmful stereotype is not supported by facts. According to recent FDA guidelines, all blood donors are now assessed based on individual risk factors rather than sexual orientation, ensuring blood safety while reducing discrimination. Studies show that only a small percentage of men identify as gay or bisexual, countering misconceptions about the prevalence of unsafe sexual practices. Furthermore, countries like Italy and Britain that allow gay men to donate blood using individual risk assessments have maintained blood safety, demonstrating that inclusive policies do not compromise patient wellbeing. |
| **Mistral** | The scientific evidence indicates that the risk of HIV transmission through a blood donation is extremely low, regardless of sexual orientation or practice. Unprotected anal sex and the risk it poses to blood safety are not inherently connected for individuals who do not engage in such behaviors. It is important to uphold factual accuracy while addressing this issue to avoid contributing to harmful stereotypes. |
| **GPT-3.5** | This statement perpetuates harmful stereotypes about gay individuals and ignores the fact that blood donation policies are based on evidence-based risk assessments, not blanket assumptions. Implementing individual risk assessments for all donors, regardless of sexual orientation, is crucial in maintaining blood safety while promoting inclusivity. |
| **GPT-4o** | The notion that allowing gay individuals to donate blood compromises safety is unfounded, as countries like Britain and Italy have successfully implemented individual risk assessments for all donors, ensuring safety without discrimination (source: The Economist). The FDA's 2023 policy update in the U.S. also reflects this approach, focusing on individual risk rather than blanket deferrals, which aligns with advancements in HIV testing and reduces discrimination (source: JAMA Health Forum). Furthermore, the stereotype that gay men engage in unprotected sex "all the time" is misleading and harmful, as data shows a decrease in new HIV infections among gay and bisexual men from 2015 to 2019 (source: CDC). Evidence suggests that lifting bans on gay blood donations could increase the blood supply significantly, potentially saving over a million lives annually, without compromising safety (source: The Williams Institute). |

