# OpenReview forum: "Fact-based Counter Narrative Generation to Combat Hate Speech"
_ACM.org/TheWebConf/2025/Conference — WWW 2025 Poster_

### Official Review · Reviewer_PQmV · 2024-11-25

**Novelty:** 3
**Technical Quality:** 3

**Review:**

Summary:
The manuscript titled "Fact-based Counter Narrative Generation to Combat Hate Speech" presents a method for generating counter narratives (CNs) to combat online hate speech by leveraging Retrieval-Augmented Generation (RAG) techniques. The authors propose integrating relevant background knowledge from multiple sources to enhance the factuality and effectiveness of CNs. The paper includes an experimental evaluation of the proposed method against a couple of baselines and discusses the results in terms of toxicity, factuality, persuasiveness, informativeness, and linguistic quality.

Pros:
1.The paper addresses a critical and timely issue of online hate speech and offers a systematic approach to generating fact-based counter narratives, which is a valuable contribution to the field.
2.The integration of external knowledge sources to enhance counter narratives is a thoughtful approach that has the potential to improve the quality of generated responses.
3.The manuscript is well-organized, and the authors have made an effort to comprehensively evaluate their method using multiple metrics, including human evaluation.

Cons:
1.The paper's primary weakness is the perceived lack of innovation. The proposed method appears to be an application of the RAG paradigm to the task of counter narrative generation without introducing distinctive design elements or novel technical contributions that significantly differentiate it from existing work in other field.
2.The paper's experimental section is limited in scope as it only compares the proposed method against two baselines from 2021. Several works mentioned in the related works are not included in the comparative analysis, which raises questions about the thoroughness of the evaluation and the generalizability of the findings.
3.The paper lacks ablation studies that would demonstrate the effectiveness of each component of the proposed framework. Without such studies, it is challenging to ascertain the individual contributions of the document retrieval, summarization, and critique model components to the overall performance.
4.The validity of the experimental metrics used to evaluate the counter narratives is questionable. While the paper employs a range of metrics, including human evaluation, the lack of clarity regarding the evaluation criteria and the demographics and background of the human evaluators undermines the reliability of these assessments.

**Questions:**

See the above “cons”.

**Reviewer Confidence:**

4: The reviewer is certain that the evaluation is correct and very familiar with the relevant literature

**Scope:**

3: The work is somewhat relevant to the Web and to the track, and is of narrow interest to a sub-community

---

### Official Review · Reviewer_j2Ja · 2024-12-02

**Novelty:** 5
**Technical Quality:** 5

**Review:**

This submission is an interesting approach to countering online hate speech by automatically generating counter narratives, which would otherwise be time/resource intensive. The methods are interesting but the evaluations are limited to measures that do not extend to 'real world' evaluations (even with the human evaluations which were not as convincing as they could be) and the methods do not introduce any clearly novel methods, but the work seems like an appropriate use of language models for designing interventions online.

Strengths of the manuscript include the clear comparison with existing methods, the breadth of the performance measures and the attempt to make them more useful in the real world, the generally clear writing, and the appropriate use of generative AI.

Abstract: The abstract could be improved with less background and  more information about the measures, dataset, and experiments, though it was nice to see a selection of quantified results.

Introduction: The statements in the start of the introduction could be improved by making them less grandiose and just sticking to the facts in terms of the positive aspects of social media.

Methods: The "human evaluation" isn't fully specified. Who are the 17 researchers? What was the variance/agreement among independent reviewers? If there was substantial variation it would be hard to claim that the evaluation was a useful one.

Results: Across all results, I could not find any information about whether the differences were significant or meaningful.

Results: The component-wise evaluation seemed less compelling, perhaps because it was hard to tell how the measures related to the actual generated text and what its potential impact might be.

**Questions:**

In social psychology, there are some nice papers showing that counter narratives are potentially useful for third parties in conversations - that we aren't just trying to influence the user who posted the original post but everyone who may see the post and the response. Could the authors add more detail about how the proposed methods could be used as tools in the real world and what would be needed to get there?

Could the authors show whether the difference in the performance across the methods were significant or meaningful? If both were put into practice, would there be a difference in the end? How much difference might it make in the real world?

Beyond the diversity and 'generic-ness', were there any investigations to try and understand *why* the performance is different from the others?

Could the authors provide more detail about the agreement in scores among reviewers? How were differences resolved?

**Ethics Review Description:**

I did not see any statement about ethics approvals for the dataset collection or an explanation of any waiver

**Ethics Review Flag:**

Yes

**Reviewer Confidence:**

3: The reviewer is confident but not certain that the evaluation is correct

**Scope:**

4: The work is relevant to the Web and to the track, and is of broad interest to the community

---

### Official Review · Reviewer_38Uc · 2024-12-03

**Novelty:** 4
**Technical Quality:** 5

**Review:**

This paper uses a retrieval-augmented approach for counter-narrative generation. The approach has four main steps: (a) use an LLM to determine three important questions to ask for counter-narrative generation; (b) use document retrieval on a knowledge base and a web search to retrieve important information; (c) summarise these using LLMs; and (d) generate counter-narratives using LLMs.

Quality:

* Overall, the quality of the paper is quite high with regards to design decisions and evaluation.
* My major concern is with potential data leakage. It appears the addition of a web search increased factuality quite a bit. Is it possible that this is simply because the CONAN dataset is either part of the GPT-4o training set or is available to your web search component?
* You could almost lose all of the introduction up to the paragraph ‘Counter narrative as a promising combat strategy’ and the paper would still be just as understanding, plus a short paragraph motivating why it is important.
* Introduction: “To enhance the quality of CNs, we propose a framework that responds with non-aggressive and fact-based feedback, by incorporating relevant background knowledge from two distinct sources.” – I think this could be expanded a bit (by removing some of the more generic text earlier in the introduction).  At this point, the reader has no idea of what this framework does (is it just a RAG-based method for example?), what the two sources are, etc. The introductory paragraph in Section 3 could pretty much be moved to the introduction. No need to keep us wondering 😊
* Many of the metrics used are not defined: BERTScore, ROUGE-L, BLEU, leaving the reader with very little information to determine whether these are useful measures and whether the results are meaningful.
* The human evaluation is quite weak. Only three samples were chosen, and only the authors’ model was used – there were no baseline comparisons or any other form of controls.

Clarity:

* I like the examples throughout the appendix.
* Throughout the related work, author names are used to identify work, but in the experiment description, model names are used; e.g. “TKGCN”. Align these better; e.g. Chung et al. [7] proposed TKGCN, which integrates generate models …”

Originality:

* As a semi-lay  reader, it is not clear to me what the difference between TKGCN and the proposed model. Table 1 is a nice overview of the models, however, the only difference appears to be the use of web search?

Significance:

* The paper does not really give much exploration for why the method works so much better than baseliens, particularly TKGCN. Is it just because there is a web search involved? Could be just add a web-search to TKGCN and get similar results? From a scientific perspective, this is what is the most important. Some ablation studies would be great here. Space could be made by shortening the introduction and by perhaps removing Section 4.6.2, which contains interesting results, but these are not as important , scientifically, as explaining why the method is so effective.

**Questions:**

Why use document summarisation instead of using information retrieval against paragraphs in the document? This would allow more targeted evidence.

Is there a chance that the Multi-Target CONAN is part of the GPT-4o training dataset or publicly available on the web for the web-search component to access? If not, how did you verify this?

Am I correct that the main difference between TKGCN and the authors’ method is that TKGCN doesn’t do a web search? Does this account for the results in factuality?

**Reviewer Confidence:**

2: The reviewer is willing to defend the evaluation, but it is likely that the reviewer did not understand parts of the paper

**Scope:**

3: The work is somewhat relevant to the Web and to the track, and is of narrow interest to a sub-community

---

### Official Review · Reviewer_w3Co · 2024-12-03

**Novelty:** 6
**Technical Quality:** 5

**Review:**

#### *Overall Assessment*

The authors present an innovative approach to counter-narrative generation to combat hate speech on social media platforms. This is an important and timely contribution as online hate speech continues to proliferate, posing significant societal challenges. The proposed framework incorporates a RAG pipeline to enhance the factuality, informativeness, and persuasiveness of counter-narratives. The authors also conduct a comprehensive evaluation, including metrics such as factuality and persuasiveness, human assessments, and comparisons to other baselines.

### *1. Quality*

•⁠  ⁠*Strengths*:
- The evaluation framework is comprehensive, assessing factuality, persuasiveness, and informativeness alongside human evaluations.
- Experimental results demonstrate clear improvements over competitive baselines, with significant gains in factuality and persuasiveness metrics.

•⁠  ⁠*Areas for Improvement*:
1. *Evalaution Robustness*: The paper does not explore the robustness of its evaluation pipeline. Specifically, factuality scores are heavily dependent on GPT-based evaluations. Including sensitivity analysis with alternative LLMs would strengthen the claims.
2. *Handling the case when no documents are retrieved*: The handling of edge cases, such as when the document filtering step results in an empty set, is not addressed. Proposing a fallback mechanism would improve the reliability of the pipeline.
3. *Use of commercial LLM*: The exclusive reliance on a proprietary model (GPT-4o) raises concerns about reproducibility. Exploring open-source alternatives or acknowledging this limitation would enhance the work.

### *2. Clarity*

•⁠  ⁠*Strengths*:
- The abstract and introduction are well-written, clearly outlining the problem and contributions of the work.

•⁠  ⁠*Areas for Improvement*:
1. *Ambiguity between Sections 4.6.1 and 4.6.2*: I personally did not understand the difference between the results shown in Sections 4.6.1 and 4.6.2

### *3. Originality*

•⁠  ⁠*Strengths*:
- The evaluation pipeline, particularly the inclusion of factuality and persuasiveness, is a significant contribution that sets this work apart from prior methods.

•⁠  ⁠*Areas for Improvement*:
1. *Comparison to related work*: The paper does not sufficiently compare its approach to [1], a closely related study on polite, fact-based counter-narratives. A detailed comparative analysis would better situate the paper's contributions.


### *4. Significance*

•⁠  ⁠*Strengths*:
- The paper addresses an urgent and important societal challenge, aligning well with the conference's focus on leveraging computational methods to mitigate online harms.


[1]: “He, B., Ahamad, M., & Kumar, S. (2023, April). Reinforcement learning-based counter-misinformation response generation: a case study of COVID-19 vaccine misinformation. In Proceedings of the ACM Web Conference 2023 (pp. 2698-2709).” which is strictly related to your topic area.

**Questions:**

Please carefully read the general review and respond accordingly.
1. How would the factuality and persuasiveness scores change if alternative LLMs were used for evaluation? Could you include experiments to test robustness across models?
2. What measures can be implemented to address cases where the document filtering step results in an empty set of relevant documents?
3. Why the method from [1] cannot be used as a baseline in your setup?

**Reviewer Confidence:**

4: The reviewer is certain that the evaluation is correct and very familiar with the relevant literature

**Scope:**

4: The work is relevant to the Web and to the track, and is of broad interest to the community

---

### Official Review · Reviewer_gZHi · 2024-12-04

**Novelty:** 5
**Technical Quality:** 5

**Review:**

**summary**:

This work proposes a fact-based counter narrative generation framework to combat hate speech. The proposed framework leverages LLMs to fact-based counter narratives. The proposed framework is a RAG framework that firstly retrieves information from both knowledge repository and online sources. The generation is then performed based on filtered, most relevant document summarization. The extensive and systematic evaluation results suggest the effectiveness of the proposed method

**Strength**:

-	The work is easy to follow
-	The proposed method is interesting and technically solid
-	The experiments are extensive, and the results seem to be promising
-	Human evaluation is included

**Weakness**:

-	The proposed framework involves multiple rounds of prompting, which may introduce additional processing time for a query. It would be better to include efficiency studies to evaluate the cost of the proposed framework.
-	More baselines are needed, especially RAG-based ones.

**Questions:**

Can you add more RAG-based baselines?

**Reviewer Confidence:**

3: The reviewer is confident but not certain that the evaluation is correct

**Scope:**

4: The work is relevant to the Web and to the track, and is of broad interest to the community